# Metabolic Potential of Microbial Communities in the Hypersaline Sediments of the Bonneville Salt Flats

Julia M. McGonigle,[a] Jeremiah A. Bernau,[b] Brenda B. Bowen,[b,c] William J. Brazelton[a,c]

aSchool of Biological Sciences, University of Utah, Salt Lake City, UT, USA
bDepartment of Geology and Geophysics, University of Utah, Salt Lake City, UT, USA
cGlobal Change and Sustainability Center, University of Utah, Salt Lake City, UT, USA

**ABSTRACT** The Bonneville Salt Flats (BSF) appear to be entirely desolate when viewed from above, but they host rich microbial communities just below the surface salt crust. In this study, we investigated the metabolic potential of the BSF microbial ecosystem. The predicted and measured metabolic activities provide new insights into the ecosystem functions of evaporite landscapes and are an important analog for potential subsurface microbial ecosystems on ancient and modern Mars. Hypersaline and evaporite systems have been investigated previously as astrobiological analogs for Mars and other salty celestial bodies, but these studies have generally focused on aquatic systems and cultivation-dependent approaches. Here, we present an ecosystem-level examination of metabolic pathways within the shallow subsurface of evaporites. We detected aerobic and anaerobic respiration as well as methanogenesis in BSF sediments. Metagenome-assembled genomes of diverse bacteria and archaea encode a remarkable diversity of metabolic pathways, including those associated with carbon fixation, carbon monoxide oxidation, acetogenesis, methanogenesis, sulfide oxidation, denitrification, and nitrogen fixation. These results demonstrate the potential for multiple energy sources and metabolic pathways in BSF and highlight the possibility for vibrant microbial ecosystems in the shallow subsurface of evaporites.

**IMPORTANCE** The Bonneville Salt Flats is a unique ecosystem created from 10,000 years of desiccation and serves as an important natural laboratory for the investigation of the habitability of salty, halite, and gypsum-rich environments. Here, we show that gypsum-rich mineral deposits host a surprising diversity of organisms and appear to play a key role in stimulating the microbial cycling of sulfur and nitrogen compounds. This work highlights how diverse microbial communities within the shallow subsurface sediments are capable of maintaining an active and sustainable ecosystem, even though the surface salt crust appears to be completely devoid of life.

**KEYWORDS** habitability, extremophiles, hypersaline, subsurface

Evaporite deposits are valuable analogs for investigating the potential for ancient and modern life in the saline conditions of extraterrestrial settings, including on Mars (1). Jezero Crater, the landing site for the Mars Perseverance Rover, contains sedimentary deposits that are expected to be the remnants of an ancient, now desiccated lake (2, 3). Gale Crater (explored by the Mars Curiosity Rover) also has vast evaporite deposits, including gypsum (4, 5). Similarly, at Bonneville Salt Flats (BSF) in Utah (UT), USA, a persistent salt pan covers ~100 km² (6, 7). Regionally, surficial lacustrine sediments are remnants of Lake Bonneville, the largest lake in western North America, before it began to desiccate ~15,000 years ago (8). Just below BSF's surface halite (NaCl) crust, layers of interbedded halite and gypsum ($CaSO_4 \cdot 2H_2O$)-rich sediments that are up to 1.5 m thick overlay carbonate muds.

Address correspondence to William J. Brazelton, william.brazelton@utah.edu.

The authors declare no conflict of interest.

Despite the sterile surface appearance of BSF, the shallow subsurface hosts remarkably robust microbial communities. These are visible as colorful pigments just below the top layer of the halite crust, and signs of life are observed within and upon the evaporite sediments with microscopy (9). However, the role of the microbial ecosystem within the geochemical and mineralogical evolution of the salt flats is generally unknown. The archaeal and bacterial taxa composition varies with sediment depth and mineralogy but is remarkably consistent across a horizontal transect of several kilometers (10). Similar microbial diversity results have been reported for hypersaline sediments in the nearby Pilot Valley basin, another remnant of Lake Bonneville, which has also been studied as a Mars analog environment due to its deposits of perchlorate (11, 12).

The seasonally limited availability of water may stimulate the density and diversity of BSF microbial communities. The salt flats are typically flooded during the winter and spring months and then rapidly desiccate during the summer (6). Surface halite dissolves with flooding and crystallizes as BSF desiccates. Gypsum, some of which crystallizes *in situ*, and some of which is detrital, is concentrated into layers via halite dissolution. When BSF's surface is desiccated, groundwater levels range from within 10 cm of the surface in the summer at BSF's center to tens of centimeters below the surface in cooler months. This annual cycle of flooding, evaporation, and desiccation contrasts with the desert salt flats of Death Valley (13), Tunisia (14), the Arabian Peninsula (15), and the hyperarid soils of the Atacama Desert (16–18), where flooding events are rare.

Microbial diversity studies of these arid to hyperarid, hypersaline systems have confirmed the presence of resident microbial communities with both cultivation-dependent and cultivation-independent studies (10, 19, 20). A few studies have investigated these systems for their abilities to support specific metabolic activities, such as carbon monoxide oxidation (20, 21), perchlorate reduction (12), and hypolithic photosynthesis (22). However, the metabolic activity, nutrient cycling, and genomic content of these microbial communities have not been studied at the level of the whole ecosystem. The role of evaporite landscapes in carbon cycling and the fluxes between land-atmosphere interfaces is unknown and is important for understanding the implications of ongoing landscape changes with declining water levels and increasing salt flats in saline lake systems worldwide (23). Here, we report our exploration of the metabolic potential of the BSF microbial community via incubation experiments and metagenome sequencing to further our understanding of ecosystem functioning in hypersaline systems.

## RESULTS

**Stable isotope incubation experiments.** The capacity for BSF sediments to support respiration was measured as the generation of $^{13}$C-labeled carbon dioxide from $^{13}$C-labeled glucose or acetate during the incubation of sediment samples at room temperature for 60 days. Our experiment was designed as a binary activity assay and was not conducive to calculating metabolic rates or a precise quantity of carbon released per gram of sediment or per unit of time. We observed the production of $^{13}CO_2$ from both glucose and acetate in all layers from the four locations included in the incubation experiments (Table 1).

Additionally, methanogenesis in BSF sediments was measured during replicate incubations as the generation of $^{13}$C-labeled methane from $^{13}$C-labeled bicarbonate, glucose, or acetate. The production of methane from bicarbonate was primarily detected in deeper sediments. The exception to this is that from surface halite samples at site 12B, where one replicate in experiments both with and without hydrogen generated $^{13}$C-labeled methane. Evidence of methane production from carbon originating in glucose or acetate was only seen in aerobic incubations from the lower horizons sampled at sites 12B and 56 (Table 1). At site 12B, methanogenesis occurred mostly in the gypsum aerobic incubations but also occurred in anaerobic incubations from the next deepest layer. Site 56 had methane production only in aerobic incubations, while site 33 had methane production only in anaerobic incubations. Similarly, site 41 had

**TABLE 1** Headspace results for presence/absence (indicated by +/− respectively) of $^{13}C$ conversions in aerobic and anaerobic incubations[a]

| $^{13}C$ conversion | Site 41 | | Site 33 | | | Site 56 | | | | Site 12B | | |
|---|---|---|---|---|---|---|---|---|---|---|---|---|
| | Layer 1 | Layer 2 | Layer 1 | Layer 2 | Layer 3 | Layer 1 | Layer 2 | Layer 3 | Layer 4 | Layer 1 | Layer 2 | Layer 3 |
| **Aerobic** | | | | | | | | | | | | |
| Glucose -> $CO_2$ | + | + | + | + | + | + | + | + | + | + | + | + |
| Glucose (-> $CO_2$?) -> $CH_4$ | − | − | − | − | − | − | + | − | − | − | + | − |
| Acetate -> $CO_2$ | + | + | + | + | + | + | + | + | + | + | + | + |
| Acetate -> $CH_4$ | − | − | − | − | − | − | − | − | − | na | + | − |
| $CO_2$ -> $CH_4$ (with $H_2$) | − − − | − − − | − − − | − − − | − − − | − − − | − − − | − − − | + + + | + − | + + | − − − |
| $CO_2$ -> $CH_4$ (no $H_2$) | − − − | − − − | − − | − − − | − − − | − − − | + − − | − − − | + + − | + − − | + + | + − − |
| **Anaerobic** | | | | | | | | | | | | |
| Glucose -> $CO_2$ | + | + | + | + | + | + | + | + | na | + | + | + |
| Glucose (-> $CO_2$?) -> $CH_4$ | − | − | − | − | − | − | − | − | na | − | − | − |
| Glucose -> $CO_2$ | + | + | + | + | + | + | + | + | na | + | + | + |
| Acetate -> $CH_4$ | − | − | − | − | − | − | − | − | na | − | − | − |
| $CO_2$ -> $CH_4$ (with $H_2$) | − − − | − − − | − − − | + + + | + + + | − − − | − − − | − − − | na | − − − | + − | + + + |
| $CO_2$ -> $CH_4$ (no $H_2$) | − − − | + − − | − − − | + + + | + + + | − − − | − − − | − − − | na | − − − | − − − | + + + |

[a]Multiple +/− values indicate replicate treatments; na indicates data is not available.

methane production in anaerobic incubations; however, here, it was observed only in incubations without added hydrogen.

**BSF metagenome-assembled genomes (MAGs).** A summary of the metagenome sequencing and assembly results is reported in Data Set S1. Sequencing results for the 16 metagenomic libraries were broadly similar across sites and sediment layers, ranging from 25 to 55 million paired reads per library. Assemblies included 700,000 to 1.6 million predicted proteins, approximately 30% of which could be annotated with the Kyoto Encyclopedia of Genes and Genomes (KEGG) database. Sample-specific assemblies were pooled for binning purposes, resulting in 43 MAGs with <15% contamination and >45% completion after automated binning. Of those, the 28 with the most complete pathways involved in biogeochemical cycles are reported here (Fig. 1; Data Set S1).

**Carbon fixation pathways.** We found evidence for multiple carbon fixation pathways in the BSF metagenomes, including the reductive pentose phosphate pathway and the Wood-Ljungdahl pathway (Fig. 2 and 3; Fig. S2). We also found evidence for

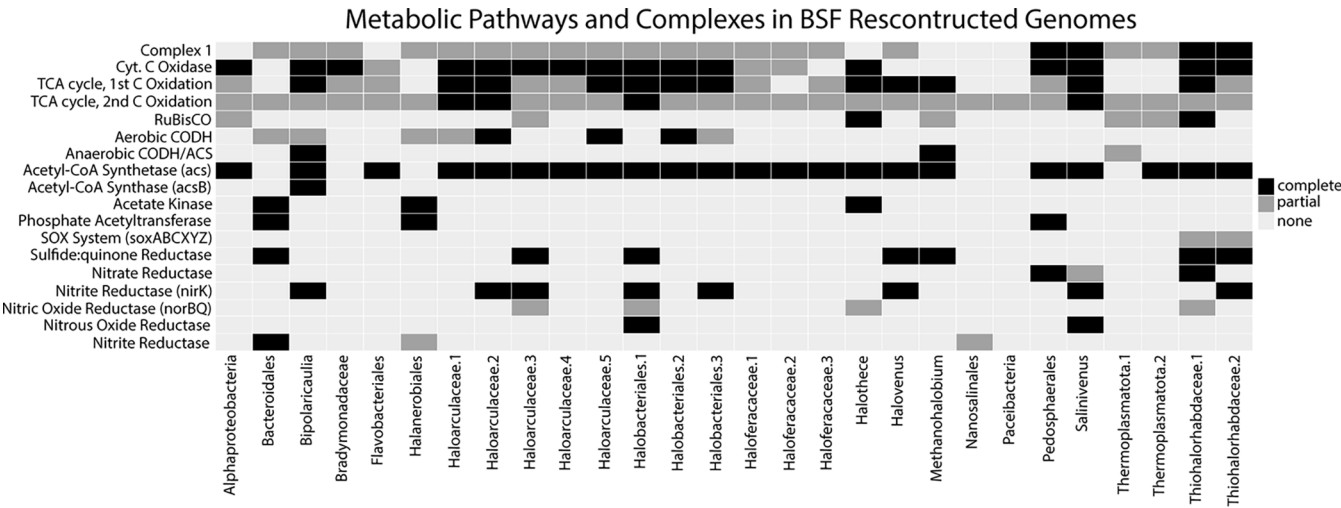

**FIG 1** Completion of various metabolic pathways and complexes in the BSF MAGs. Black indicates a complete pathway/complex, dark gray indicates a partial pathway/complex, and light gray indicates an absence of any genes in the pathway/complex. Pathways/complexes correspond to the following KEGG modules and IDs: Complex I (M00144), Cyt. c oxidase (M00155), TCA cycle, 1st C oxidation (M00010), TCA cycle, 2nd C oxidation (M00011), RuBisCO (K01601, K01602), aerobic CODH (K03518, K03519, K03520), anaerobic CODH/ACS (K00194, K00197, K00198), acetate kinase (K00925), phosphate acetyltransferase (K00625), sulfide:quinone reductase (SQR), nitrate reductase (napAB or narGHI), and nitrous oxide reductase (nosZ).

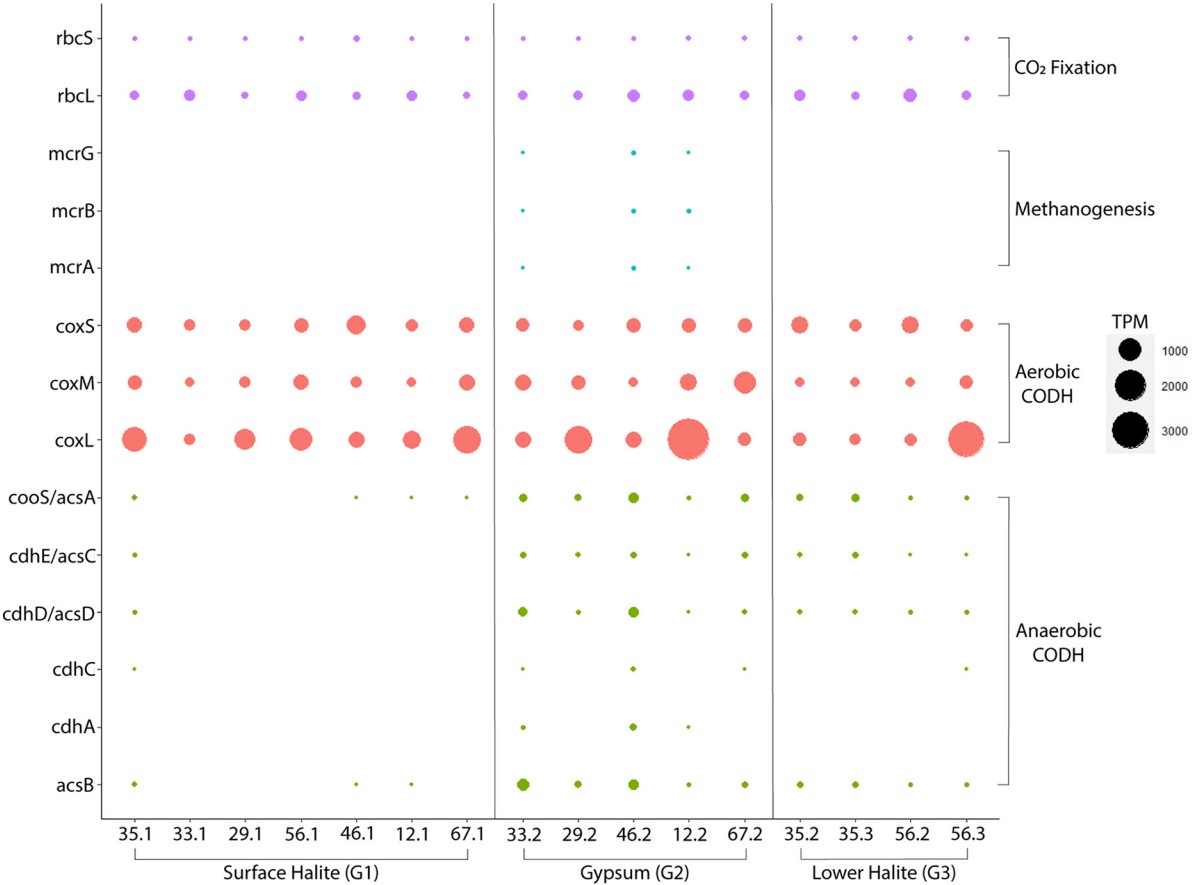

**FIG 2** Abundance of key carbon cycling genes in BSF metagenomes (values in transcripts per million [TPM]).

carbon monoxide oxidation, which can serve as a source of carbon for autotrophy and/ or supplemental electrons for energy conservation.

The reductive pentose phosphate pathway (i.e., the Benson-Calvin cycle, RPP) features two key enzymes: ribulose-1,5-bisphosphate carboxylase/oxygenase (RuBisCO) and phosphoribulokinase. Both small and large subunits are distributed across all layers, but the large subunit is generally more abundant (Fig. 2). The small subunit of RuBisCO appears to be heterogeneously distributed across all samples, but the large subunit is slightly more abundant, with coverage greater than 144 transcripts per million (TPM) in the surface halite crust at site 33 and in lower layers from sites 46, 12B, 56, and 35 (Fig. 2; Data Set S2). A complete KEGG module for the RPP pathway is found in all of our metagenomes. Genes encoding both the small and large subunits of RuBisCO in addition to phosphoribulokinase are present in only two of our MAGs: the cyanobacteria Halothece, which contains a complete RPP pathway, and Thiohalorhabdaceae.1, which is only missing fructose-1,6-bisphosphatase (Fig. 1). The large subunit is found in numerous MAGs, but many are missing the small subunit and phosphoribulokinase, another key enzyme in the RPP.

A key enzyme of the Wood-Ljungdahl (WL) pathway is anaerobic CO dehydrogenase/ acetyl-CoA synthase (CODH/ACS), a five-subunit complex. The genes *acsCD* encode the corrinoid iron-sulfur protein component of the complex responsible for reactions in the methyl branch of the WL pathway, and acsAB encode the catalytic component responsible for $CO_2$ reduction and acetyl-coA synthesis. Although not abundant in most samples, these genes are present in our metagenomes, mostly from the gypsum and lower halite layers (Fig. 2). The metagenome constructed from the gypsum layer at site 46 is the only sample with a complete suite of CODH/ACS genes. Most of these genes have the highest abundance in this sample. The gypsum layer metagenome from site 33 has a comparably high abundance

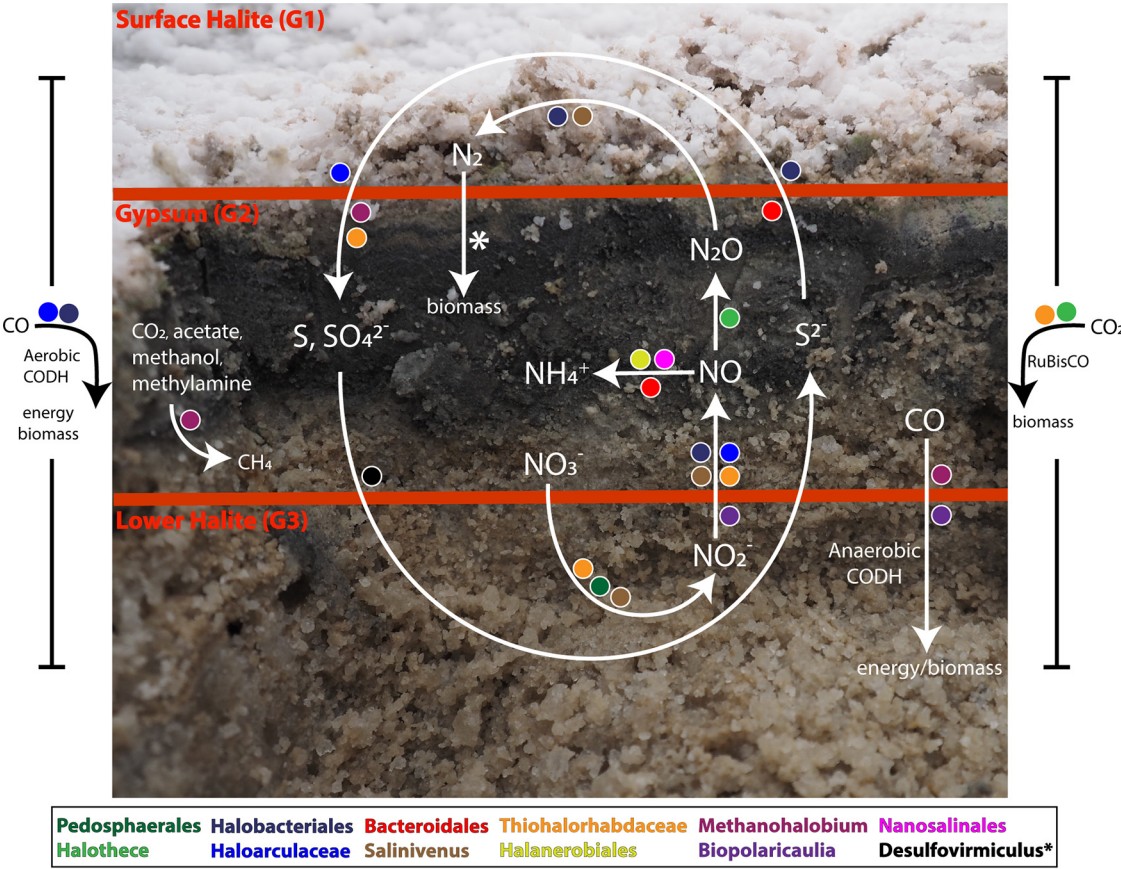

**FIG 3** Overview of metabolic pathways predicted to be active in BSF sediment groups and associated taxa. *, gene was not recovered from a MAG, and taxa were inferred via gene homology (Data Set S1 contain BLAST results). Note that this graphic represents an artistic generalization of the data. In some cases, the difference in gene abundance is minor between sediment groups. Refer to Fig. 2 and 5, Fig. S4, and Data Set S2 for precise values.

of anaerobic CODH/ACS genes and is only missing *cdhB* (the epsilon subunit) (Data Set S2). All of the metagenomes constructed from the gypsum and lower halite layers contain a complete KEGG module for the WL pathway. In contrast, only one metagenome from the surface halite samples contained a complete module for the pathway (site 35). The others are missing methyltransferase (*acsE*) and *acsCD*. Additionally, a commonly used marker gene for the WL pathway, acetyl-CoA synthase (*acsB*), was found in the surface halite only at sites 12B, 46, and 35, but it was present in all of the metagenomes from the gypsum and lower halite layers.

Some MAGs captured a partial anaerobic CODH/ACS complex, but only the *Bipolaricaulia* MAG contained genes encoding all crucial parts of the complex, including both catalytic units (Fig. 1). This *Bipolaricaulia* MAG also contained the most complete WL pathway, only missing the genes encoding formate dehydrogenase. The *Thermoplasmatota*.1 MAG contained the bacterial form of the catalytic subunits of CODH/ACS, but it was missing additional parts of the complex. This MAG contained only two other genes in the WL pathway: methylenetetrahydrofolate dehydrogenase (*folD*) and formate-tetrahydrofolate ligase (*fhs*). This may be due to the incomplete representation of the genome in our data, as the *Thermoplasmatota*.1 MAG is only 52% complete (Data Set S1).

**(i) Methanogenesis.** Genes for methanogenesis were not abundant across the BSF samples, except for *mttB*, which encodes a trimethylamine methyltransferase used in methanogenesis from trimethylamines (Fig. S2, Data Set S2). In all of the metagenomes, there was an absence of methylenetetrahydromethanopterin dehydrogenase (*hmd*), which is present in some methanogens that are reliant on hydrogen gas as an

electron donor (24). A complete suite of genes for methanogenesis from carbon dioxide (with the exception of *hmd*) was found in only two of the BSF samples, namely, the gypsum layers from sites 33 and 46. In addition, the metagenome from gypsum at site 12B had a nearly complete pathway for methanogenesis from acetate, methylamines, and methanol, with only *mtrCDEH* (tetrahydromethanopterin S-methyltransferase) and *mtd* (methylenetetrahydromethanopterin dehydrogenase) missing. The key enzyme for methanogenesis, methyl-coenzyme M reductase (*mcr*), was not found in any other metagenomes (Fig. 2).

We obtained only one MAG, classified as genus *Methanohalobium*, with a nearly complete pathway for methanogenesis. The MAG contained the genes required for methanogenesis from $CO_2$, methylamines, methanol, and acetate, but it did not include any encoding hydrogenases.

**(ii) CO oxidation.** The enzyme aerobic carbon monoxide dehydrogenase (CODH) enables the use of carbon monoxide (CO) as an energy and/or carbon source in oxic conditions. The three genes encoding aerobic CODH (*coxSML*) were among the most abundant genes in all sediment layers (Fig. S2). The gene encoding the catalytic subunit (*coxL*) was particularly abundant in surface halite at sites 35 and 67B but was found at the highest abundance in gypsum at site 12B (Fig. 2). In the lower halite layers, *coxL* was most abundant at site 56. 8 of the 28 MAGs obtained contain at least 1 of the 3 aerobic CODH genes (Fig. 1), but the catalytic large subunit was only found in bins belonging to haloarchaea. The bacterial *Bacteroidales* and *Bipolaricaulia* bins both contained genes encoding the small subunit, and the *Haloanerobiales* bin contained genes encoding the small and medium subunits.

The genes encoding form I CODH typically appear in an operon as *coxMSL*, while form II CODH genes are either organized as *coxSLM* or are not contained in an operon (25). The complete form I CODH operon is found in our *Haloarculaceae*.5 and *Haloarculaceae*.2 MAGs (Data Set S1). These two MAGs plus a third (*Haloarculaceae*.2) have additional sets of genes annotated as *coxSLM* or *coxLM*, matching the operon structure of form II CODH. However, these genes lack the AYRGAGR active site motif characteristic of previously reported form II CODH genes (25). Interestingly, none of our MAGs containing an aerobic CODH included additional genes with annotations associated with carbon fixation.

CO oxidation by *Archaea* has recently been described with isolates cultured from BSF (20, 21). We calculated the average nucleotide identity (ANI) among our haloarchaea bins and the available genomes for these CO-oxidizing *Archaea* (Data Set S1). The best match between the three recently characterized isolates and our MAGs was 79.93% ANI (for *Halovenus carboxidivorans*) (Data Set S1), indicating that our MAGs probably represent distinct species or genera compared to the previously described CO-oxidizing species (20, 21).

**Nitrogen fixation.** Genomic evidence for nitrogen fixation was present in four of the seven sites (12B, 67B, 33, and 35). Site 12B had the highest abundance of nitrogenase genes (*nifDHK*) (Data Set S2). Unfortunately, no nitrogenase genes were present in a high-quality MAG, so we investigated individual nitrogenase sequences present in the metagenomes by querying them against the NCBI nr database. Six nitrogenase sequences from the surface halite sample collected at site 12B had high sequence similarity (~55% to 75% amino acid identities) to *nifDH* genes previously identified in bacterial *Chloroflexi* (*Roseiflexus*) species. However, all of the nifK genes found in this sample had the highest sequence similarity to metagenomic sequences representing an unclassified archaeon (Data Set S1). In the lower sediment layers, the nitrogenase genes are most similar to those previously described in purple nonsulfur bacteria or in sulfate-reducing bacteria (*Deltaproteobacteria*). Sites 12B and 35 had one *nifH* and *nifK* gene (respectively) with sequence similarity to *Halorhodospira* (purple sulfur bacteria) species.

**(i) Nitrification.** Hydroxylamine dehydrogenase (*hao*), which is involved in the first step of nitrification, was detected only in the lower halite metagenomes from sites 35 and 56 (Data Set S2). Evidence for other genes associated with nitrification is limited. Genes annotated as methane/ammonia monooxygenase (*pmoBC/amoBC*) were also

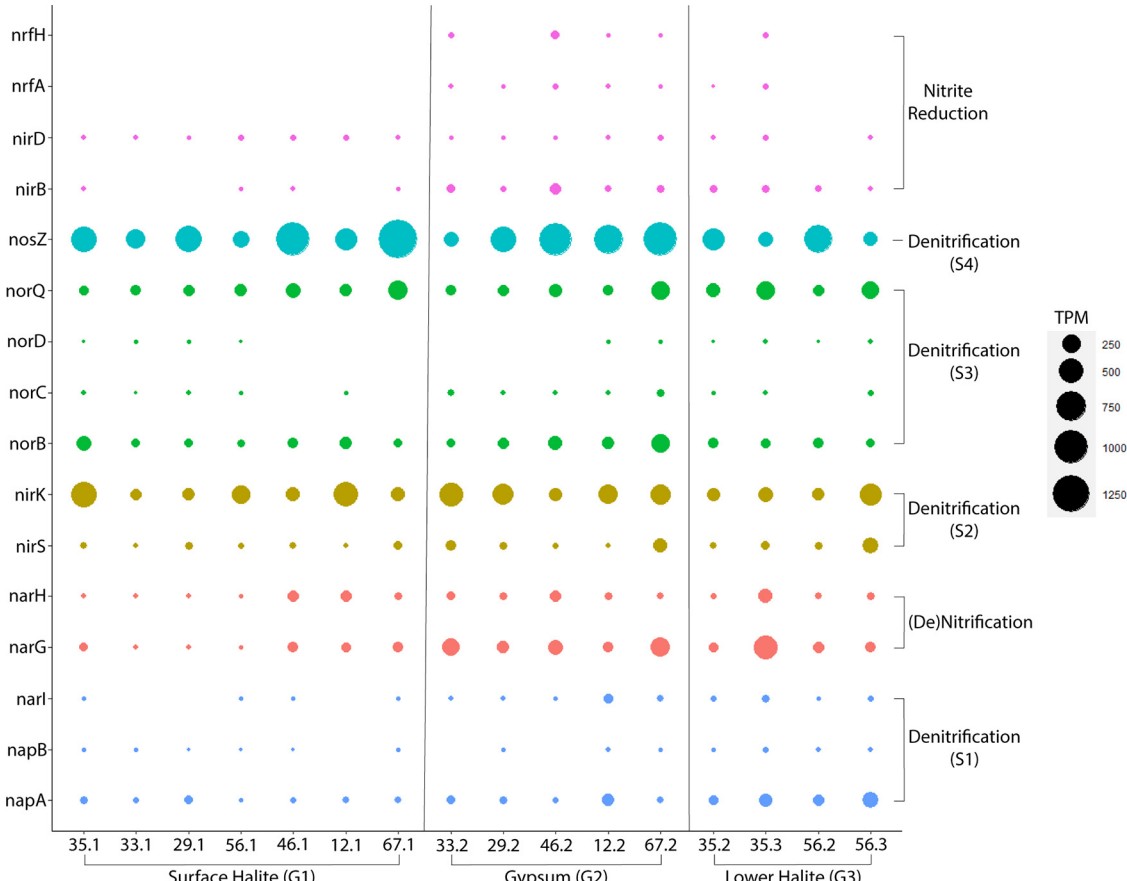

**FIG 4** Abundance of nitrogen cycling genes in BSF metagenomes (values in transcripts per million [TPM]). The nifDHK, hao, and pmo/amoBC genes were removed from this plot for readability, as the abundance and processes are discussed in the text and are insignificant in the overall BSF nitrogen cycle.

detected at low levels in the deeper layers of sites 35 and 56. Still, additional evidence is required to distinguish whether these genes are involved in the oxidation of methane or ammonia, as KEGG orthology alone is not sufficient. These genes were only found in one MAG (*Thiohalorhabdaceae*.2), which is 59% complete and lacked the alpha subunit (Data Set S1).

KEGG orthology alone cannot distinguish nitrite oxidoreductase (*nxr*; involved in the nitrification of nitrite to nitrate) from nitrate reductase (*nar*; the reverse reaction involved in denitrification). Genes annotated as the alpha and beta subunits of these enzymes (*nxrAB/narGH*) were present in all samples and were highly abundant in the upper gypsum and lower halite layers (Fig. 4). Again, the only MAG to contain both of these genes represents a *Thiohalorhabdaceae* (Fig. S3), and this MAG also contains the gamma subunit of the enzyme specific to nitrate reductase (*narI*). The type species of this group is *Thiohalorhabdus denitrificans*, an extremely halophilic chemolithoautotroph that is capable of denitrification but not nitrification (26).

**(ii) Denitrification.** Denitrification is a complex microbial process that can be broken into four steps carried out by one or multiple organisms (Fig. 3). The first reaction in the process (nitrate to nitrite) overlaps with complete nitrate reduction, and the genes *narGH* are challenging to distinguish from the *nxrAB* that is involved in the reverse reaction, nitrification (discussed above). Nitrate reductase catalyzes this step in denitrification, with two distinct dissimilatory types: *napA* represents a periplasmic version of the enzyme, and *narGH* represents the respiratory version (27). These three nitrate reductase genes were present in all samples but never at a high abundance, compared to other nitrogen-cycling enzymes (Data Set S2). While the respiratory version seems more

abundant than the periplasmic type (under the assumption that all annotated narGH are nitrate reductase), both types appear to be the most abundant in the lower halite sediment layers (Fig. 4). Only three MAGs contained genes that were involved in this first step (Fig. 3). The *Salinivenus* and *Pedosphaerales* MAGs contained the periplasmic version of nitrate reductase (*napA*), and, as previously mentioned, the *Thiohalorhabdaceae*.1 MAG contained *narGHI* (Fig. 1).

Two genes involved in the second step of denitrification (*nirSK*; nitrite reductase) are widespread and abundant across BSF (Data Set S2). While *nirK* has a fairly even and high abundance pattern throughout the sediment layers, *nirS* was generally much less abundant, except at sites 56 and 67B (Fig. 4). No MAGs contained *nirS* annotations, but *nirK* was found in eight MAGs (one *Salinivenus*, two *Halobacteriales*, three *Haloarculaceae*, one *Bipolaricaulia*, and one *Thiohalorhabdaceae*.2).

Genes encoding nitric oxide reductase (*norBC*) catalyze the third step of denitrification. These genes were only present at low abundances, but *norB* was widespread across BSF (Fig. 4). A functional nitric oxide reductase is believed to require the accessory proteins norQD (28). In the BSF metagenomes, *norQ* was as widespread and abundant as *norB*. The only MAG that included catalytic nitric oxide reductase genes was Halothece, which contained norB but was missing the *norQD* accessory protein genes. Three MAGs contained *norQ* genes but not *norB*: *Halobacteriales*.1, *Haloarculaceae*.3, and *Thiohalorhabdaceae*.1 (Fig. 1).

The last step in denitrification is carried out by nitrous oxide reductase, encoded by the gene *nosZ*. This gene is the most abundant nitrogen cycling gene at BSF and is relatively evenly distributed among sediment layers (Fig. 4). Curiously, it has a roughly inverse abundance distribution with *nirK* (i.e., samples with a lower abundance of *nosZ* have a higher abundance of *nirK*). The *Salinivenus* and *Halobacteriales*.1 MAGs are the only two MAGs that included *nosZ*.

**(iii) Dissimilatory nitrate reduction to ammonia.** The last step in dissimilatory nitrate reduction to ammonia is mediated by nitrite reductase. Genes encoding the large subunit of the NADH-dependent form of the enzyme (*nirB*) were less abundant or were absent from the metagenomes constructed from the surface halite layer (Fig. 4). Genes encoding the cytochrome c form of the enzyme (*nrfAH*) were completely absent from all surface halite and all sediment layers at site 56. Generally, both *nir* and *nrf* genes are more abundant in the gypsum layers. Three MAGs contained genes encoding nitrite reductase (Fig. 1). The *Halanerobiales* MAG contained only the large subunit of the NADH-dependent form of the enzyme, whereas the *Nanosalinales* MAG only contained the small subunit (*nirD*). The Bacteroidales MAG contained both genes encoding the cytochrome c form of nitrite reductase (*nrfAH*).

**Sulfur cycle genes.** The gene encoding sulfide-quinone reductase (*sqr*) is among the most abundant genes in our BSF metagenomes (Fig. S2 and S3; Data Set S2), and it was present in seven MAGs (*Halobacteriales*.1, *Haloarculaceae*.3, *Haloarculaceae*.5, *Bacteroidales*, *Thiohalorhadbaceae*.1, *Thiohalorhadbaceae*.2, and *Methanohalobium*) and all of the metagenomes (Fig. 1).

Other genes involved in the oxidation and reduction of sulfur species are widespread in BSF sediments but are generally present at low abundances (Fig. 5, Data Set S2). Genes encoding enzymes involved in dissimilatory sulfate reduction (*dsrAB*, *aprAB*) are found at most sites below the surface halite crust, but they are most abundant in the gypsum from sites 33 and 46 (Fig. 3 and 5). Unfortunately, no MAGs contained *dsrAB* genes, so we investigated individual *dsrAB* genes from the metagenomes by querying them against the NCBI nr database (Data Set S1). A total of 11 *dsrA* and 14 *dsrB* genes had high sequence similarity to *Desulfovermiculus halophilus*, and these were abundant in the gypsum layers in our previous survey of 16S rRNA genes (10). Other notable BLAST hits were to uncultured taxa sequenced from other salty environments, including the Great Salt Lake.

Genes encoding the SOX system, associated with thiosulfate oxidation, were most abundant in the gypsum at site 12B and were included in two MAGs (*Thiohalorhadbaceae*.1 and

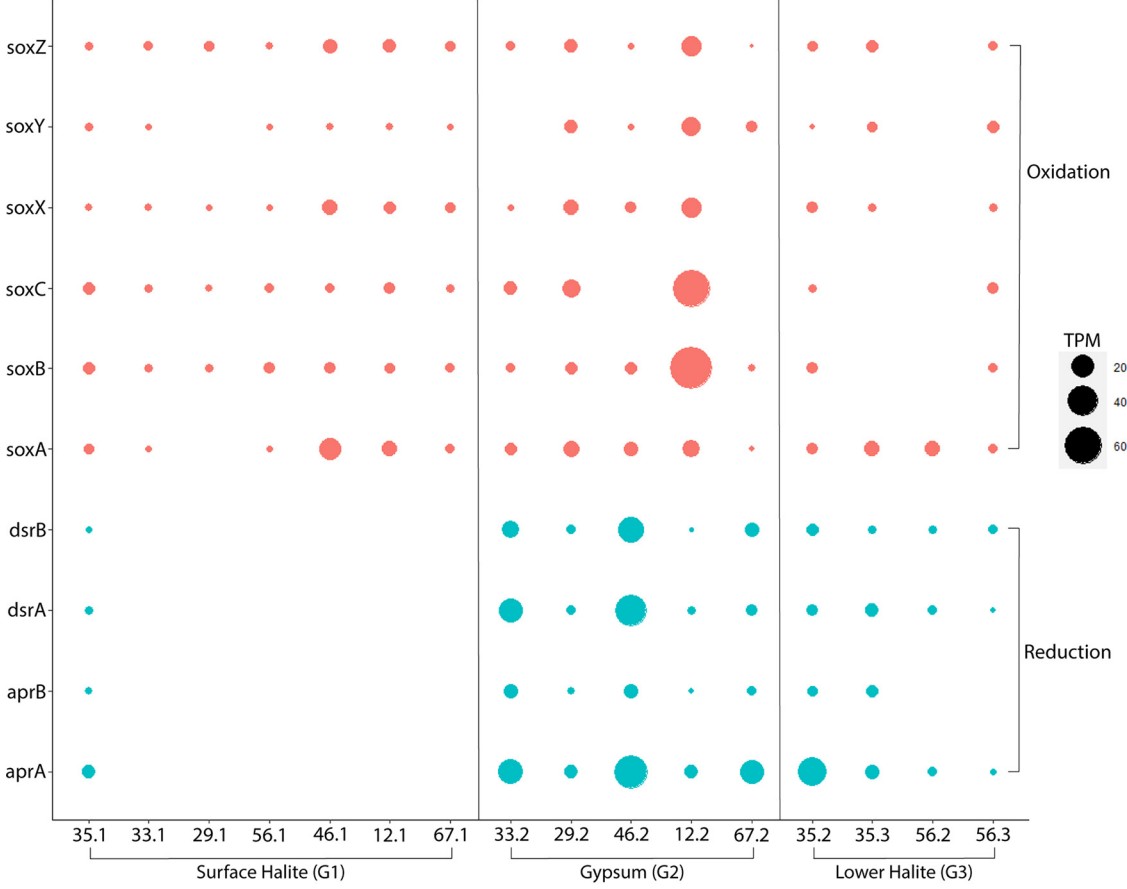

**FIG 5** Abundance of sulfur cycling genes in BSF metagenomes (values in transcripts per million [TPM]). The SQR gene is removed for readability. The high abundance of this gene compared to the other BSF metabolic genes is shown in Fig. S2. The sat gene is also removed for readability, as this gene is highly abundant and is involved in sulfur assimilation and reduction.

*Thiohalorhadbaceae.*2). However, these MAGs were missing the catalytic component (SoxABC) and only included predicted sequences for SoxYZ. The genes associated with sulfur oxidation and sulfur reduction generally had opposite abundance patterns; samples in which SOX genes were more abundant had a lower abundance of genes involved in both dissimilatory and assimilatory sulfate reduction (Fig. 5).

## DISCUSSION

**BSF has an active microbial ecosystem.** Even though the Bonneville Salt Flats (BSF) appear to be utterly desolate at the surface, the layers of sediment just below the surface host a diverse, abundant, and active microbial ecosystem (Fig. 3). The quantity of DNA extracted from these samples indicates the presence of dense microbial communities comparable to those of soil communities (10), although some portion of the extracted DNA is likely to be extracellular DNA that is preserved within the sediment and is not from live organisms. As we have previously reported, the microbial community composition exhibited remarkably low variation among the sampling sites, but the new results presented here reveal that within each site, the abundance of key metabolic genes shifted from the surface halite crust to the gypsum and lower halite layers (Fig. 2–5). In particular, genes associated with methanogenesis, acetogenesis, denitrification, and sulfur reduction were more abundant in the gypsum layer, compared to the surface halite crust (Fig. S4). Furthermore, our incubation experiments demonstrated the potential for both aerobic and anaerobic respiration in all sediment layers, and methanogenesis occurred during incubations of deeper sediment layers (Table 1).

**Carbon fixation occurs at BSF via multiple pathways.** As expected, the dominant carbon fixation pathway at BSF is the reductive pentose phosphate (RPP) pathway, and

genes encoding the RuBisCO enzyme are widespread at roughly similar abundances across all locations and sediment layers (Fig. 2; Fig. S4). Cyanobacteria and algae (*Dunaliella salina*) are known to be important primary producers in salty ecosystems, such as BSF (29). Indeed, we found evidence for *Halothece*-like cyanobacterial species with a complete RPP pathway. The only other MAG containing a complete RPP pathway was classified as *Thiohalorhabdaceae* (Fig. 1). An autotrophic lifestyle and the presence of RuBisCO have been previously described for the type genus for this family, *Thiohalorhabdus* (30). Therefore, these taxa likely represent one of the primary producers at BSF.

Our previous survey of microbial diversity at BSF identified 16S rRNA gene sequences classified as putative acetogenic bacteria (*Acetothermia*). Indeed, the most complete Wood-Ljungdahl (WL) pathway was found in a MAG assigned to the class *Bipolaricaulia* (within the phylum *Bipolaricaulota*, which was previously referred to as *Acetothermia* and candidate division OP1) (31, 32). This MAG appears to account for most of the CODH genes detected in the lower sediment layers and it may be one of the primary producers in the anaerobic zones of BSF. While the *Bipolaricaulia* MAG contained a nearly complete WL pathway, it lacked genes encoding an acetate kinase and a phosphate acetyltransferase, which are typically required for an acetogenic lifestyle (Fig. 1). Both of these enzymes are also absent from 14 publicly available *Bipolaricaulota* MAGs that encode the WL pathway (33). Additional analyses of these MAGs are required to assess their potential for autotrophic growth on $CO_2$, heterotrophic homoacetogenic fermentation, and syntrophic growth via acetate oxidation, as proposed for other *Bipolaricaulota* (33). Nevertheless, the presence of a nearly complete WL pathway in the *Bipolaricaulia* MAG reported here suggests the possibility of acetogenesis as an important component of the BSF microbial ecosystem.

**Methanogens at BSF have a diverse suite of potential carbon sources.** The BSF metagenomes contain an incredible amount of metabolic flexibility for methanogenesis, with genes enabling the use of carbon dioxide, acetate, methanol, and mono-, di-, and tri-methylamines. The results from our incubation experiments and genomic analyses indicate that the methanogens at BSF do not require hydrogen as an electron source. Methane production occurred in the absence of hydrogen, and sequences encoding hydrogenases associated with methanogenesis were absent.

All of the methanogenesis genes were included in one MAG, which was classified as *Methanohalobium*. This genus is within the order *Methanosarcinales*, which are known to use a variety of substrates for methanogenesis (34). The ability to use methylated carbon sources in methanogenesis is a known strategy by which to overcome competition for $CO_2$ in saline environments (35, 36). Furthermore, *Methanohalobium* belongs to a group of methanogens that tend to be more tolerant of oxidizing conditions, which could explain the methane production observed in surface crusts (37).

The production of $^{13}CH_4$ from $^{13}C$-glucose in our aerobic incubations is evidence of heterotrophic carbon turnover, as glucose cannot be converted directly to methane by any known species (Table 1). We captured higher methane production under aerobic conditions, and no methane production occurred in anaerobic glucose incubations, even though these experiments captured conversion to $^{13}CO_2$. In support of the hypothesis that aerobic heterotrophy aids methanogenesis, *mcr* genes are only found in gypsum sediments, just under the surface halite, where limited oxygen is likely to be present in pores. Curiously, acetate-supported methane production occurred in only one sample (the site 12B gypsum layer), suggesting that other organisms might outcompete methanogens for acetate at BSF. Unfortunately, the metagenomic sequencing of the deeper samples (halite mixed with gypsum; mineralogical group 4) was unsuccessful. Additional sampling might capture methanogenesis genes in deeper sediments.

**Haloarchaeal CO-oxidizers at BSF expand the diversity of known carboxydovores.** Genes predicted to enable the aerobic oxidation of CO are widespread throughout most locations and sediment layers in the BSF (Fig. 2; Fig. S4). Interestingly, all but one of our MAGs (*Halanerobiales*) that encode aerobic CO dehydrogenase are classified

as *Haloarchaea* (Fig. 1). This group, whose CO-oxidizing ability has only recently been described, is also known to use the light-driven proton pump bacteriorhodopsin for supplemental energy (20). It is likely that these BSF haloarchaea represent carboxydovores, which are unable to grow solely on CO and use atmospheric concentrations of CO as supplemental energy (38). Multiple mechanisms (e.g., CO oxidation, light-harvesting complexes) by which to capture supplemental energy could be extremely beneficial for these heterotrophs living in extreme environments that impose additional costs for osmoregulation.

Known CO-oxidizing haloarchaea have previously been isolated from salty environments, including BSF (20). Our haloarchaea MAGs are distinct from the three BSF isolates (*Halobacterium bonnevillei*, *Halobaculum saliterrae*, and *Halovenus carboxidivorans*) that have been previously reported, and they are most likely new species or genera of haloarchaea (average nucleotide identities [ANI] of <80%). Furthermore, our results indicate that these novel CO-oxidizing haloarchaea encode form I aerobic CODH. These MAGs have additional genes that are annotated as aerobic CODH but lack significant sequence similarity with any characterized forms of CODH. Further work should explore whether these sequences represent members of form II CODH, a distinct form of CODH, or another enzyme entirely.

The only nonarchaeal MAG (*Halanerobiales*) containing aerobic CODH genes is missing *coxL*, which encodes the catalytic subunit of the enzyme. A functional enzyme might still be present in this species because the *coxL* gene family is diverse and is known to be frequently misannotated (25). Nevertheless, the absence of bacterial *coxL* in these metagenomes suggests that CO oxidation in BSF may be carried out only by archaea.

**Denitrification is the center of the BSF nitrogen cycle.** Denitrification is the most widespread and abundant portion of the BSF nitrogen cycle (Fig. S2, Fig. 4). The taxonomy of MAGs containing denitrification genes has been previously described as highly abundant organisms at BSF through 16S rRNA analyses (e.g., *Salinivenus*, *Halobacteriales*, *Nanosalinales*, *Halanerobiales*, and *Thiohalorhabdaceae*) (10, 11). While numerous MAGs contained genes for denitrification, none contained genes for all four steps in the process (Fig. 1). Two MAGs (*Salinivenus* and *Halobacteriales*) encode multiple steps of denitrification. Curiously, the *Salinivenus* MAG has three genes involved in the process: *napA* (step 1), *nirK* (step 2), and *nosZ* (step 4). Atypical *nosZ* genes, encoding a functional complex, are previously described in the type species for the *Salinibacteraceae* family, *Salinibacter ruber* (*S. ruber*) (39). However, culture work shows that *S. ruber* is unable to grow with nitrate as an electron acceptor, and denitrifying species in this family have not been identified (40).

Interestingly, we found *nosZ* to be a highly abundant gene (Fig. S2). This is in apparent contrast to other environmental studies, which have shown that species containing *nosZ* are often less abundant than microbes with genes encoding the preceding reductases in the denitrification pathway, such as *nirK* (41). It is to be noted that while *nosZ* is more abundant overall, the BSF samples with the highest abundances of *nirK* seem to have a lower abundance of *nosZ* (Data Set S2).

Denitrifying haloarchaea have been described, but it is unclear whether they participate in partial or complete denitrification (42). Our MAGs suggest that the BSF haloarchaea are involved in partial denitrification, particularly in reducing nitrite to nitric oxide (*nirSK*). A functional NirK enzyme has been extracted from a cultured member of *Halobacteriales* (*Haloferax mediterranei*) (43). Interestingly, the same haloarchaea MAGs have aerobic CODH, and denitrification linked with CO oxidation has been previously reported for bacteria (44).

The abundance of genes associated with denitrification contrasts sharply with the sparse distribution of nitrogen fixation genes. Nitrogenase-encoding sequences were relatively rare or were absent in all of the BSF metagenomes. For example, *Halothece* cyanobacteria are commonly known as nitrogen fixers (45). Yet, the only nitrogen-associated gene in our *Halothece* MAG was *norB*, which catalyzes the reduction of nitric

oxide during denitrification. The few nitrogenase sequences in BSF had similarities with those encoded by photoautotrophic and/or sulfur-cycling bacteria. The lack of nitrogenase sequences compared to the diversity and abundance of denitrification genes suggests that the BSF microbial community has sufficient access to fixed forms of nitrogen throughout the sediment layers sampled during this study.

**BSF contains a dynamic sulfur cycle with surprising players.** Genes associated with dissimilatory sulfate reduction are concentrated in the gypsum layer at each site. Unlike halite, which dominates both the surface crust and the deeper layers at most sites, gypsum provides an important electron acceptor for microbial metabolism (i.e., sulfate). Furthermore, sulfate is readily available as an aqueous solute in BSF brines (Data Set S2). Therefore, the concentration of gypsum in a layer near the surface but protected from UV radiation, desiccation, and higher salinity appears to provide a foundation for a dynamic sulfur cycle, in which the reduction of sulfate minerals provides reduced sulfur compounds as an energy source for a variety of microbial species (Fig. 3).

Metabolic pathways for sulfide oxidation are predicted to be abundant and widespread across BSF, including the surface halite crust. The oxidation of sulfur compounds may occur through aerobic or anaerobic pathways, using alternative electron acceptors, such as those used in denitrification (46). The gene encoding sulfide-quinone reductase (SQR) is one of the most abundant genes in the BSF metagenomes. This enzyme can be involved in energy conservation, by linking sulfide oxidation to ATP production, or in sulfide detoxification, in which the oxidation of sulfide only serves to prevent damage to the cell (47–50).

We identified diverse taxa encoding SQR, including *Thiohalorhadbaceae*, *Halobacteriales*, *Haloarculaceae*, *Bacteroidales*, and *Methanohalobium*. The SQR gene is not typically associated with haloarchaea or methanogens, but it has been identified in a *Bacteroidales*-like species through metagenomic studies of a Siberian soda lake (51). Interestingly, the *Bacteroidales* MAG has the genomic capacity to couple sulfide oxidation to nitrite reduction, thereby linking the two macronutrient cycles. Supporting this, the genome lacks a cytochrome c oxidase, along with other genes typically associated with aerobic metabolism. It does contain *nrfA*, which has been shown to use nitrite as a respiratory electron acceptor in *Escherichia coli* (52).

Our results indicate that *Thiohalorhadbaceae*-like species are important players in the BSF sulfur cycle. They likely play a role in sulfite (and possibly sulfate) assimilation for amino acid synthesis and sulfide oxidation for energy conservation (via SQR). Even though the type species for this family, *Thiohalorhabdus denitrificans*, is known to oxidize thiosulfate (26), neither of our *Thiohalorhadbaceae* MAGs contain catalytic *sox* genes. While other *sox* genes are present in the metagenomes, the *Thiohalorhadbaceae* MAGs only contain *soxYZ* genes, which were recently shown to encode a protein that acts as only a carrier in the multistep process of thiosulfate oxidation (53). This is consistent with the absence of *soxB* genes in salt marsh sediments in which *Thiohalorhabdus*-like species were also described (54). Not much is known about this largely uncultured family, but these results suggest either that some members of *Thiohalorhadbaceae* are unable to oxidize thiosulfate or that they do so with novel enzymes.

The inclusion of SQR in our *Methanohalobium* MAG is consistent with its presence in the genome of *Methanolobus* species, who belong to the same family of methanogens (*Methanosarcinaceae*) and are known to be capable of sulfide oxidation (55). Metagenomic sequences predicted to encode SQR have also been associated with the archaeal methanotrophs ANME-1, who have been proposed to be capable of using SQR in the reverse direction to produce sulfide instead of or in addition to oxidizing sulfide (56, 57). Environmental surveys indicate that SQR sequences are diverse and are mostly uncharacterized, indicating a potentially widespread but unknown role in sulfur-rich environments (47).

**Conclusions.** This metagenomic study has produced an inventory of potential energy sources and metabolic pathways that are feasible in the evaporite sediments of the Bonneville Salt Flats (BSF). Furthermore, laboratory incubation experiments

confirmed that BSF microbial communities are capable of aerobic and anaerobic respiration and methanogenesis from multiple carbon sources. Microbial activity appears to be most diverse and active within a subsurface, gypsum-rich layer, but the biogeochemical cycling of carbon, nitrogen, and sulfur involves organisms throughout the sediment profile, including multiple layers of halite and gypsum (Fig. 3). Reduced sulfur and nitrate are a likely redox couple for multiple species of BSF archaea and bacteria. In particular, genomes of haloarchaea (*Halobacteriales* and *Haloarculaceae*) encode the potential for sulfide oxidation, CO oxidation, and denitrification, suggesting that they may play a central role in the cycling of carbon, nitrogen, and sulfur in BSF (Fig. 3).

As an ecosystem that has persisted in a hypersaline and rapidly changing environment, BSF is an important natural laboratory for investigating the past, present, and future potential for life on Mars. The diverse microbial communities of BSF are capable of maintaining an active and sustainable ecosystem, even though the surface salt crust appears to be completely devoid of life. The seasonal cycle of flooding and desiccation at BSF is likely to be an important driver of microbial structure and activity. Future work should investigate the temporal dynamics of microbial activity at BSF in comparison with systems that experience more permanent desiccation. In addition, gypsum deposits appear to play a key role in stimulating microbial activity within BSF sediments, as evidenced by the surprising diversity of organisms predicted to metabolize sulfur compounds and metabolic genes within gypsum-rich sediments. Mars contains expansive evaporite deposits, including vast deposits of gypsum in Gale Crater and in the north polar region, and recent missions have begun to explore their underlying shallow subsurface environments (4, 5). Future work should investigate whether the gypsum-oriented microbial community described here is dependent on oxygen or whether a similar community could thrive in anaerobic conditions, such as those likely to have been present in the subsurface of Mars. If hypersaline sediments on Mars have not been habitable in the past or present, then the further study of BSF and similar environments on Earth will be important for understanding which requirements for life have been lacking on Mars.

## MATERIALS AND METHODS

**Sampling and DNA extraction.** Sampling locations, elemental analyses, and DNA extraction methods were previously described in detail. Briefly, sediment samples were collected from the upper 30 cm of sediments at 8 sites distributed across BSF in September of 2016 (Fig. S1). Sediment samples from seven of these sites yielded successful metagenomes (Data Set S1), and four representative sites were chosen for incubation experiments. Environmental measurements of temperature, pH, conductivity, and dissolved oxygen for the brine at each site are provided in Data Set S2. No rain had occurred at the site during the summer prior to the sample collection in September of 2016.

At each site, distinct sediment layers were sampled via the visual identification of color and textural characteristics. Sediment samples were grouped according to their mineralogy (group 1, surface halite; group 2, gypsum; group 3, lower halite; and group 4, halite mixed with gypsum) as described in detail in (10). Briefly, surface halite samples (mineralogical group 1) consist of halite crystals, minor gypsum (2.5% to 27%), and trace amounts (<0.5%) of clay. These are relatively enriched in chloride, potassium, and zinc (from most to least abundant). The gypsum samples (mineralogical group 2) consist of layered fine to medium-sized gypsum grains and are composed of 29% to 39% gypsum and 6% to 22% halite. The gypsum samples contain the largest amount of clay and detrital minerals of any stratum (0.3 to 1.7%), and they are enriched in sulfur, calcium, magnesium, aluminum, potassium, iron, strontium, zinc, barium, and manganese. Lower halite samples (mineralogical group 3) consist of cemented halite with porous vertical dissolution pipes. These have an elemental composition similar to that of the surface halite. The gypsum layer was underneath the surface halite layer in all locations except for sites 35 and 56, where the gypsum layer was absent (Fig. S1).

Samples were transported on ice within 12 h of collection to the University of Utah, where they were stored at $-80°C$. Microbiological analyses, including DNA extraction, were performed in the following 6 months. DNA extraction was performed using a protocol from (58) that was modified for highly saline material. The modified protocol was published by (10), is archived as doi:10.5281/zenodo.2827066, and is summarized here. A sterile mortar and pestle were used to crush the collected sediments. A 0.25 g subsample was extracted using a buffer containing 0.1 M Tris, 0.1 M EDTA, 0.1 M $KH_2PO_4$, 0.8 M guanidium HCl, and 0.5% Triton X-100. The samples were then subjected to one freeze-thaw cycle, incubation at 65°C for 30 min, and beating with 0.1 mm glass beads in a Mini-Beadbeater-16 (Biospec Products) to lyse. Purification was performed via extraction with phenol-chloroform-isoamyl alcohol, precipitation in ethanol, washing in Amicon 30K Ultra centrifugal

filters, and final cleanup with 2× SPRI beads (59). DNA quantification was performed with a Qubit fluorometer (Thermo Fisher).

**Stable isotope incubation experiments.** Incubation samples were collected alongside DNA sequencing samples at sites 12B, 33, 41, and 56. Sediment (~5 g) from each layer was placed into a sterile plastic tube containing 40 mL of either oxic or anoxic 18% artificial seawater solution. The anoxic solution was made in an anoxic chamber (Coy Laboratory Products) with 0.05% dithiothreitol (DTT). Exposure to oxygen during the sampling of the sediments was unavoidable, so we only attempted to mitigate oxygen exposure rather than maintain completely anoxic conditions in the field. Instead, samples were transported (cooled but not frozen) to the University of Utah, where aerobic and anaerobic incubations were prepared on the same day as the sampling.

In the laboratory, the sediment samples were prepared for incubations to test for respiration (the conversion of $^{13}$C-labeled glucose or acetate to $^{13}CO_2$) and methanogenesis (the conversion of $^{13}$C-labeled bicarbonate, glucose, or acetate to $^{13}CH_4$). All treatments were tested in both aerobic and anaerobic conditions. In addition, half of the incubations received an injection of $H_2$ gas to test for their stimulation of microbial activity. After the sediment samples were vortexed to break apart the sediments and to create a sediment slurry, the following steps were performed in the anoxic chamber for anaerobic incubations and on the lab bench for aerobic incubations. 2 mL of the sediment slurry were pipetted into an exetainer (LabCo) tube containing 2 mL of anoxic or oxic saltwater media amended with one of the following: bicarbonate, glucose, or acetate, each at 0.07 M and labeled with 25% $^{13}$C (i.e., mixed 1:4 with $H^{13}CO_3$, $^{13}C_{12}H_6O_{12}$, or $CH_3^{13}CO_2$, and the respective $^{12}$C compound; Cambridge Isotope Laboratories). All of the saltwater media were adapted from the Halohandbook (60), contained NaCl, $MgCl_2·6 H_2O$, $MgSO_4·7 H_2O$, KCl, TrisHCl, and included B vitamins (thiamine and biotin) as well as the following trace elements: $MnCl_2·4 H_2O$, $ZnSO_4·7 H_2O$, $FeSO_4·7 H_2O$, and $CuSO_4·5 H_2O$. Around 10 mL of hydrogen gas was added to half of the bicarbonate incubations using a syringe. The incubations were conducted at room temperature in either light conditions on a lab bench (aerobic incubation) or dark conditions inside an anoxic chamber (anaerobic incubations) for 60 days. After this time, around 7 mL of the headspace from each exetainer tube were sampled with a syringe and transferred into two clean exetainers flushed with nitrogen gas in the anoxic chamber. Exetainers containing the sampled headspace were sent to the UC Davis Stable Isotope Facility for measurements of $^{13}CH_4$ and $^{13}CO_2$ (Thermo Scientific GasBench system interfaced to a Thermo Scientific Delta V Plus isotope ratio mass spectrometer). The production of methane from one of the provided $^{13}$C-labeled sources was inferred to have occurred in the experiments in which the $\delta^{13}$C of headspace $CH_4$ was highly enriched (+4 to +2000%). No methane production from the provided $^{13}$C-labeled sources was inferred to have occurred in the experiments in which the $\delta^{13}$C of $CH_4$ was $-8$ to $-46$%. The production of carbon dioxide from one of the provided $^{13}$C-labeled sources was inferred to have occurred in the experiments in which the $\delta^{13}$C of the headspace $CO_2$ was highly enriched (+18 to +27,000% VPDB).

**Metagenome sequencing and analysis.** The metagenomic DNA was fragmented to ~500 to 700 bp using a Qsonica Q800R sonicator. According to the manufacturer's instructions, 500 ng of fragmented DNA were used to construct metagenome libraries using the NEBNext Ultra DNA Library Preparation Kit for Illumina. The University of Utah High-Throughput Genomics Core Facility performed the quality control and the sequencing of the metagenomic libraries. The quality was evaluated with a Bioanalyzer DNA 1000 chip (Agilent Technologies), and then paired-end sequencing (2 × 125 bp) was performed on an Illumina HiSeq2500 platform with HiSeq (v4) chemistry. The 24 metagenome libraries were multiplexed with 8 libraries per 1 Illumina lane, yielding a total of 845 million paired reads for all 24 libraries before quality filtering. The demultiplexing and conversion of the raw sequencing base-call data were performed through the CASAVA (v1.8) pipeline.

The raw sequences were processed by the Brazelton lab to trim the adapter sequences with BBDuk (part of the BBTools suite, v35.85 [61]) in order to remove artificial replicates and to trim the reads based on quality. The removal of replicates and the quality trimming were performed with our seq-qc package (https://github.com/Brazelton-Lab/seq-qc). At this stage, 8 of the libraries were determined to be of low quality and were not included in further analyses. The remaining 16 libraries had 600 million paired reads after quality filtering. Contigs were assembled from the paired-end reads with MegaHit v1.1.1 (62), using k-mers of 27 to 141. Prodigal v2.6.3 (63) was run in the anonymous gene prediction mode to identify open reading frames. Functional annotation was performed using the BLASTP function of Diamond v0.9.14 (64) with both the prokaryotes and the T10000 (addendum annotations) databases from KEGG release 83.2 with an E value of 1E−6. Annotations were selected by the highest quality alignment, as determined by the bit score. Coverage was determined through read mapping with bowtie2 v2.3.2 (65) and bedtools v2.25.0 (66).

**Metagenome binning.** The binning of individual samples resulted in the poor representation of diverse taxa identified through previous 16S rRNA studies and recovered quality genomes of only a few taxa. Because our previous work showed a striking similarity in microbial community composition along a horizontal transect (10), we pooled contigs from multiple assemblies to improve the metagenomic binning efforts. Metagenomic binning was performed on the pooled contigs using BinSanity (67). The bins were evaluated using CheckM (68), and taxonomy was assigned using GTDB-Tk Classify (kb_gtdbtk v.0.1.4) through Kbase (69). As with the unbinned contigs, functional annotation was performed using the BLASTP function of Diamond v0.9.14 (64) with both the prokaryotes and the T10000 (addendum annotations) databases from KEGG release 83.2. The online KEGG mapper tool was used to identify bins with more complete metabolic pathways of interest. ANI was

performed with fastANI v0.1.2 (https://github.com/ParBLiSS/FastANI) through the online data platform Kbase.

**Data availability.** All unassembled sequences related to this study are available at the NCBI Sequence Read Archive (BioProject accession number PRJNA522308). All of the MAGs may be found under NCBI BioSample accession numbers SAMN19270115 to SAMN19270142. All supplemental material, solution recipes, and protocols are archived at https://doi.org/10.5281/zenodo.5569980. All custom software and scripts are available at https://github.com/Brazelton-Lab.

## SUPPLEMENTAL MATERIAL

Supplemental material is available online only.

**DATA SET S1**, XLSX file, 0.2 MB.

**DATA SET S2**, XLSX file, 0.05 MB.

**FIG S1**, PDF file, 0.2 MB.

**FIG S2**, PDF file, 0.3 MB.

**FIG S3**, PDF file, 0.6 MB.

**FIG S4**, PDF file, 0.8 MB.

## ACKNOWLEDGMENTS

We gratefully acknowledge Betsy Kleba and Emily Dart for their critical roles in collecting samples for this study.

No competing financial interests exist.

This research was supported by the National Science Foundation Award 1617473 (CNH-L: Adaptation, Mitigation, and Biophysical Feedbacks in the Changing Bonneville Salt Flats), the NASA Astrobiology Institute Rock-Powered Life Team, and the NASA Postdoctoral Program (NPP).

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
