## [Reviewer comments · mSystems]

Metabolic Potential of Microbial Communities in the Hypersaline Sediments of the Bonneville Salt Flats

Julia McGonigle, Jeremiah Bernau, Brenda Bowen, and William Brazelton

Corresponding Author(s): William Brazelton, University of Utah

Review Timeline:

Submission Date:	September 2, 2022
Editorial Decision:	September 25, 2022
Revision Received:	October 21, 2022
Accepted:	October 24, 2022

Editor: Sean Gibbons

Reviewer(s): The reviewers have opted to remain anonymous.

Transaction Report:

DOI: <https://doi.org/10.1128/msystems.00846-22>

Re: mSystems00373-22 (Metabolic Potential of Microbial Communities in the Hypersaline Sediments of the Bonneville Salt Flats)

Dear Dr. Julia M McGonigle:

I have received the reviews of your manuscript. While your paper addresses an interesting question, the reviewers stated several concerns about your study and did not recommend publication in mSystems without major revisions. In particular, please note the following.

As you know, at mSystems we are committed to making rapid final decisions. Because it appears that addressing the reviewers' concerns will require a significant amount of additional work that would delay the ultimate outcome, my decision at this time is to reject the manuscript.

If you feel that you wish to address the criticisms of the reviewers, you may submit a revised manuscript to mSystems as a new submission, which will be assigned a new manuscript number and receipt date. Please note the previous manuscript number and my name in the cover letter. Provide point-by-point responses to the issues raised by the reviewers in a file named "Response to Reviewers," not in your cover letter. Upload a compare copy of the manuscript (without figures) as a "Marked-Up Manuscript" file. In the response file, specify with page and line numbers where the revisions have been made in the marked-up manuscript.

Both reviewers expressed concern that your manuscript lacked a digestible synthesis of the results that gave the reader a clear take-home message. I suggest you make a substantial effort in your revision to build a clear narrative that integrates your results into a 'systems' perspective of both the biotic and abiotic components of the ecosystem.

Sincerely,

Sean Gibbons

Editor, mSystems

Reviewer comments:

Reviewer #1 (Comments for the Author):

McGonigle and colleagues present a metagenomic interrogation of microbial communities from the Bonneville Salt Flats, along with evidence that the sampled communities are active when tested in the laboratory. In general, this work is on a very interesting microbial ecosystem and is methodologically sound, and I applaud the authors on their substantial and comprehensive data and code sharing. However, I find the manuscript to be incomplete. The authors have an interesting dataset where they can answer (and I believe it is a key goal of theirs) how microbial functions change along a geochemical gradient from halite to a gypsum mix, with the astrobiology relevance of some martian soils being rich in gypsum. Yet, reading this manuscript I am left uncertain and unclear how microbial functions are changing across the different

geochemistries sampled by the authors. A rework of the text and figures, with some suggestions below, would help lend some clarity to the results and conclusions.

Our response:

We have thoroughly revised the text and figures to more clearly highlight that many of the key observations and conclusions of our study are connected to differences between the halite and gypsum layers at each site. Our original nomenclature for the samples (i.e. "group 1, group 2, group 3") apparently obscured this fundamental observation, so we have replaced this nomenclature with direct references to halite and gypsum in the text and figures. This revision, plus editing of the discussion to highlight these interpretations, should address the reviewer's concerns about a take-home message and astrobiological relevance.

Specific comments:

How long had it been since the last rain event when the samples were taken?

Our response:

Sampling occurred in early September of 2016. According to our records, it had not rained since the Spring of 2016. Note that this sampling occurred before the installation of the BFLAT weather station, so there were no reliable records of rainfall on the salt at that time. However, the racers had a successful summer 2016 season with many events, and these only occur when the salt is very dry and there is no seasonal rain. A sentence describing the lack of summer rain prior to sampling was added to the methods.

Line 38 - typo: "the the"

Our response:

The typo has been corrected.

Introduction: can the authors describe how from a geochemical perspective, the BSF is a suitable analog for the Martian environment? In particular what is known about perchlorates at BSF? In addition, in the "Importance" section they mention gypsum - but can they add what is known about the gypsum composition of martian soil to the introduction (currently only in discussion)?

Our response:

We have added a sentence describing the presence of gypsum in Gale Crater and clarified that the nearby Pilot Valley basin is rich in perchlorate.

General comment throughout the results: I think the manuscript would read far more strongly if the authors referred to each sample or group of samples by their general characteristics, instead of their arbitrary group numbers. A reader will be unfamiliar with what "group 2" or "sites 33 and 46" are without referring to the details elsewhere - instead, the manuscript should say "samples high in gypsum... samples from three sites that had geochemical properties X, Y, and Z... two sites X miles apart, etc." when reporting results throughout the manuscript.

Our response:

We have made these changes, as requested. Note that the between-site variability was very low, and nearly all of the important trends observed in this study were between sediment layers at the same site. The only references to individual sites are in the results section, where reporting of fine-grained results may be of interest to some readers.

I also think a simplified version of figure S1 could be incorporated into Figure 1 of the text to provide readers with a better understanding of which samples are which. Similarly, throughout the figures, I would HIGHLY suggest that the authors improve upon the "Group 1" "35.1" "33.1", etc. labels with more human-readable labels that lend themselves to biological interpretation.

Our response:

We have labeled the layers in Figure 3 as "surface halite", "gypsum", and "lower halite", as requested, and we have replaced the "Group 1, 2, 3" nomenclature with direct references to the minerals. The "35.1", "35.2", etc. labels are only used in the figures, where their meanings should be clear based on context, and their full mineralogical references (halite, gypsum, etc.) are used in the text.

Line 249 - A minor quibble, but I don't think it should be said that a metagenome "encodes" for a gene - microbial genomes can "encode", but with respect to a metagenome, it is best to simply say that a gene was present, or how abundant that gene is.

Our response:

We replaced "encoded" with "included" or "contained" or similar verbs when referring to metagenomes, reserving "encode" to refer to a specific protein or enzyme encoded by a gene.

Section 3.4 - Nitrogen Fixation - the presence of genes for nitrogen fixation in the unbinned fraction of the metagenome is interesting, but I think their origins can be more carefully interrogated. Instead of simply reporting the closest hit of the individual *nifH* genes, they should report the taxonomic hits of all proteins on the contigs. For example, if a *nifH* gene is found on a large contig with 50 other proteins, it can be helpful to summarize or visualize the best hits of the genes on that contig to determine its taxonomic origin. One tool to do this is CAT: <https://github.com/dutilh/CAT>. Even better, if there are any SCGs on that contig, they can further be used for direct phylogenetic inference.

Our response:

Unfortunately, none of the *nifH* sequences were on large contigs, consistent with their non-inclusion in any high-quality MAGs. Therefore, our analysis of the *nifH* genes themselves, as reported in the text, is the most information we can obtain for these sequences without additional sequencing data or a completely different computational approach that would be outside the scope of this study.

Incidentally, we had previously attempted to use CAT in our lab to obtain taxonomic classifications of individual contigs, but we encountered technical issues with storage requirements for the database and version conflicts with dependencies.

Figures 2, 4, and 5, are all too visually similar and also overly simplistic. Why split results for MAGs (Figure 1) and all genes (the rest of the figures)? Why not have a Figure for each biogeochemical cycle that incorporates data from both genes and MAGs? Further, using tools like CAT, some taxonomic resolution of the genes abundances in these figures should be provided and visualized.

Our response:

We do not know why visually similar figures should be undesirable. They provide a uniform style for interpreting the same analysis performed on different sets of genes.

Including individual gene coverage in each metagenomic sample as well as metabolic pathway composition for each MAG in a single figure is highly challenging. Nevertheless, we already attempted to do so with Figures S3a and S3b, where the widths of arrows represent coverage across all samples and the colored circles indicate which genes are encoded by each MAG. In addition, Figure 3 provides a digestible overview that highlights differences among sediment layers while acknowledging that the biogeochemical cycles require organisms in all of the layers.

We sympathize with the desire to have taxonomic context for the genes shown in Figures 2, 4, and 5, and we attempted a version of these figures that incorporates information about which genes are included in which MAGs. The resulting figure was very busy, however, and obscured the main point of the figures, which is to show changes in gene abundance between mineralogical groups. Moreover, we already report the distribution of genes among MAGs in Figures 1, 3, and S3. Finally, including taxonomic classifications from a contig classification tool such as CAT would be unnecessarily confusing because most of the genes in these figures are included in MAGs, which are more reliable for taxonomic classification.

Figure 3- I like this figure, but how does it change across the different types of samples the authors collected? Can we get a figure like this, contrasting halite vs gypsum?

Our response:

This figure indeed contrasts the surface halite crust with the underlying gypsum deposits, but this was apparently obscured by our unclear labeling of the figure. We have labeled the layers more clearly and thoroughly revised the text to highlight the low site-to-site variability but strong gradients between layers at each site.

Line 566 - "In addition, gypsum deposits appear to play a key role in stimulating microbial activity within BSF sediments" This sounds like an important conclusion, but it is lost on me how the authors reached it from their data. If they could rework the manuscript to emphasize these findings, that would be a simple and key takeaway.

Our response:

We have thoroughly revised the methods, results, discussion, and figures to highlight the many observations in this study that relate to the significance of the gypsum deposits. The original manuscript obscured these results by referring to gypsum layers as "mineralogical group 2", which we have now corrected in the revision.

Reviewer #2 (Comments for the Author):

This article is a follow-up to McGonigle et al. 2019, which explored the microbial abundance and diversity, structures, and trace elements of four distinct evaporite layers in the Bonneville Salt Flats. This manuscript follows with shotgun metagenomic sequencing in which functional genes for a variety of different biogeochemical processes are identified, with the addition of incubation experiments with labeled carbon substrates to identify heterotrophic activity and methanogenesis in each layer with and without presence of oxygen and/or hydrogen as an electron donor. The authors do a thorough analysis of important C, N, and S-cycling genes and which MAGs/metagenomes from each group harbor each genes (and the completeness or lack thereof of various metabolic pathways) and this paper would be a great addition to the field of microbial ecology in extreme, astrobiology-relevant ecosystems. However, there is surprisingly little discussion of the salt flat structure's chemical properties, waterflow, etc as they apply to the microbial processes observed in each layer, and taking the time to address this both in the introduction and discussion would greatly improve the impact of the paper, and wouldn't necessarily require major revisions. (I recognize that the authors may have been hesitant to do so without a more thorough collection of water chemistry data, but should not wait until the conclusion to discuss it).

Overall suggestions/questions:

- Introduction doesn't do justice to the research gap that this paper would fill/why research into microbial nutrient cycling in these evaporite sites is so important for both astrobiology and microbial ecology.

Our response:

The introduction has been revised to more clearly highlight the astrobiological relevance and the context of previous research.

- Describe the three groups (the three mineralogical layers) in the intro or methods, not just with reference to McGonigle 2019, (found them in lines 194 to 195 but shouldn't need to hunt for it) - and why/how you might expect the microbial metabolic functions in each layer to be different. For the purposes of clarity and easier reading, suggest that the groups be referred to more often by their mineralogical composition (upper halite, gypsum, lower halite). More discussion of the mineralogical properties of gypsum and its greater nutrient availability (as compared to the salt crusts) would provide more context for your results

Our response:

We have followed the reviewer's suggestion and referred to the sediment layers throughout the text as halite and gypsum instead of the mineralogical groups. In addition, we have more thoroughly discussed the significance of gypsum to microbial ecology and astrobiology in the revised introduction and discussion.

- There is no discussion of linking the metabolic processes/carbon incubation experiments explored in this paper to the trace chemistry done in McGonigle 2019/the data in the supplementary data set, or discussion of what the supplementary data. Maybe there is nothing of interest there, but a sentence or two explaining what a reader should take away from it would be good

Our response:

We have highlighted the brine chemistry data in our discussion of the importance of sulfate as an electron acceptor. Otherwise, these data are provided as a courtesy for the scientific community.

- Did metagenomic sequencing yield similar results to the 16S sequencing in terms of which species dominated in which layers?

Our response:

Yes, our metagenomic results largely agree with the results of the prior 16S rRNA study, and we highlighted a few examples of this in the text.

Lines 50-52

I can see this has already been mentioned in previous reviews, but the info about vehicle land speed records and potash comes out of nowhere in terms of paragraph structure and is still rather awkward. (and most people outside of Utah or speed racing aren't going to be familiar with the Bonneville Salt Flats). I would suggest ending the first paragraph with another sentence or two explaining why it's important to understand what types of microbial communities and biogeochemical cycling are present in Earth analogs of remnant lakebeds in terms of both astrobiology and terrestrial ecology, and then make paragraph 2 an introduction to BSF as a field site (in which the details about speed records etc will make more contextual sense).

Our response:

We have revised the introduction accordingly.

Line 79

Why were those four sites out of seven chosen?

Our response:

We were unable to conduct experiments for all seven sites, so we chose four of them to represent the spatial variability of the full set of seven.

Line 96-127

This section would benefit from being much clearer on which parts of the incubation experiments are testing for which questions – the results section for the incubations is a little hard to interpret. Or, focus the attention on the results section for the incubations including being clearer on the science questions you are addressing with the different types of incubations.

Response:

We have added an overview of the incubation experiments to the methods section. In addition, note that the first sentence of the results section states: "The capacity for BSF sediments to support respiration was measured as the generation of ¹³C-labeled carbon dioxide from ¹³C-labeled glucose or acetate during incubation of sediment samples at room temperature for 60 days." The first sentence of the following paragraph states: "Additionally, methanogenesis in BSF sediments was measured during replicate

incubations as the generation of ¹³C-labeled methane from ¹³C-labeled bicarbonate, glucose, or acetate."

Line 98

Description of individual layers or criteria used? How much sediment?

Our Response:

We have clarified the sentence to include the quantity (~5 g) and by changing "individual layers" to "each layer". Table 1 shows which samples and layers were included in the experiments.

Lines 124-125

"no methane production was inferred to have occurred in experiments where the $\delta^{13}\text{C}$ of CH_4 was -8 to -46 ‰" no methane production from the provided ^{13}C sources?

Our Response:

We have clarified the sentence to include "from the provided ^{13}C sources", as suggested.

Lines 144-145

Paired-end reads Contigs were assembled from paired-end reads with MegaHit v1.1.1

Our Response:

The sentence has been modified as suggested.

Lines 150-152

Were you running fastANI before or after binning? Seems unusual to run it on contigs

Our Response:

The sentence has been moved to the binning section, as suggested. Furthermore, the program's documentation states, "FastANI supports pairwise comparison of both complete and draft genome assemblies."

Lines 159 and 164

Both sentences seem to say the same thing (that taxonomy of MAGs was assigned with GTDB-Tk).

Our Response:

The redundancy has been removed in the revision.

Line 194

At least one or two sentences, in the methods or intro or results, explaining these groups further, should be added. Ideally a reader should be able to extract meaning without reading the previous paper

Our Response:

We added a paragraph to the methods section describing the layers. In addition, the text has been revised throughout to highlight the significance of the mineralogical layers.

Line 202

Why 28, specifically?

Our Response:

These 28 MAGs have the most complete metabolic pathways, as stated in the text. The others were excluded because their pathways were not complete enough to warrant discussion.

Line 393

“compared to some soil communities” – expand on this?

Our Response:

We merely refer to the similarity in biomass to soil communities, generally. We have omitted the word "some" from this sentence in the revision in an attempt to clarify this point.

Lines 445-447

“not the carbon molecules produced during anaerobic respiration” – does this refer to other molecules besides CO₂? If not, why would methanogens not use CO₂ produced under anaerobic conditions? This needs a little more explanation/support.

Our Response:

We have deleted this sentence from the revision. The stimulation of methanogenesis under aerobic conditions during our study is mysterious, and the now-deleted sentence was an attempt to provide a potential explanation. We agree, however, that the explanation was unsatisfying.

September 25, 2022

Dr. Julia M McGonigle
University of Utah
257 S 1400 E, Rm 201
Salt Lake City, UT 884112

Re: mSystems00846-22 (Metabolic Potential of Microbial Communities in the Hypersaline Sediments of the Bonneville Salt Flats)

Dear Dr. Julia M McGonigle:

Thank you for submitting your manuscript to mSystems. We have completed our review and I am pleased to inform you that, in principle, we expect to accept it for publication in mSystems. However, acceptance will not be final until you have adequately addressed the reviewer comments outlined below.

Preparing Revision Guidelines

Sincerely,

Sean Gibbons

Editor, mSystems

Journals Department
American Society for Microbiology
1752 N St., NW

Reviewer comments:

Reviewer #1 (Comments for the Author):

The authors have responded to several of my prior points well. However, I still feel like their work is missing a few critical elements that would greatly increase the scientific value of the work.

Namely, they describe differences in community metabolic capacity between the three surface halite, gypsum, and lower halite layers. These differences are elegantly displayed in Figure 3. However, Figures 2, 4, and 5 are in my opinion still insufficient. It is challenging to infer trends between the three strata from the sizes of the circles. Instead, if the authors used a "jittered" or "Swarm" boxplots, readers would be easily able to intuit the degree of differences in metabolic pathway abundances between strata. The authors could also then support the robustness of these differences with statistical test comparing these distributions between strata. For example, it looks like there is more capacity for Methanogenesis in the gypsum samples - but is this difference statistically significant? Anaerobic carbon monoxide dehydrogenase being absent from the surface halite is another example that looks promising. However it's really hard to discern these trends from the sizes of the circles along! Thus, an update to these figures along with some statistical tests would be extremely valuable.

Reviewer #2 (Comments for the Author):

This is the second review for this article, which is a follow-up to McGonigle et al. 2019 and explores the functional genes and possible metabolisms in the three uppermost layers of the salt crust in the Bonneville Salt Flats as predicted by shotgun metagenomic data, as well as the capacity for heterotrophy or methanogenesis in aerobic or anerobic (w/ or w/out H₂) conditions using incubation with stable isotope-labeled carbon sources.

Most of the comments from both reviewers have been addressed, with additions to the introduction that more comprehensively describe the importance of this study and the research gap it fills, as well as clarifying which layer is which by referring to the salt pan layers by their mineralogical properties rather than by group number.

However, the main comment that both myself and the other reviewer shared is that, while excellent and comprehensive in itself, the discussion of functional genes/MAGs isn't fully linked to the minerology of the site - i.e., the three different layers - from a systems perspective. Only a few sentences have been added to the discussion and conclusion to address this point, and while the additions are definitely good ones and provide some synthesis, I'm not sure it goes far enough. If, most of the time, there aren't any firm conclusions that can be drawn about microbial metabolic interplay between the layers (e.g. because, despite differences in abundance, most of these genes are spread throughout the layers), that is fine, but these things are not explicitly stated. (none of this would necessarily be a concern if this journal weren't specifically focused on microbes as they relate to systems). The addition of one or two sentences to other sections of the discussion, similar to the addition of the S-cycling section, and perhaps more specific wording at the beginning of the discussion (how does the abundance of key metabolic genes shift?) would help a lot.

Other minor notes

- Is organic matter as well as microbial biomass concentrated in the gypsum layer? (it certainly looks like it from the picture and from some wording in the text but it's never explicitly stated as far as I can tell)
- Might be interesting to add a sentence speculating on the possibility of how a similar microbial community might operate on a salt flat on ancient Mars, in anerobic conditions
- Noticed typos/misspellings line 53, 60.

Response to Reviewers

Reviewer #1 (Comments for the Author):

The authors have responded to several of my prior points well. However, I still feel like their work is missing a few critical elements that would greatly increase the scientific value of the work.

Namely, they describe differences in community metabolic capacity between the three surface halite, gypsum, and lower halite layers. These differences are elegantly displayed in Figure 3. However, Figures 2, 4, and 5 are in my opinion still insufficient. It is challenging to infer trends between the three strata from the sizes of the circles. Instead, if the authors used a "jittered" or "Swarm" boxplots, readers would be easily able to intuit the degree of differences in metabolic pathway abundances between strata. The authors could also then support the robustness of these differences with statistical test comparing these distributions between strata. For example, it looks like there is more capacity for Methanogenesis in the gypsum samples - but is this difference statistically significant? Anaerobic carbon monoxide dehydrogenase being absent from the surface halite is another example that looks promising. However it's really hard to discern these trends from the sizes of the circles along! Thus, an update to these figures along with some statistical tests would be extremely valuable.

Our Response:

We constructed jitter plots as alternatives to Figures 2, 4, and 5, as requested. We still prefer the bubble plots as a more aesthetically pleasing and richer visualization of the dataset, but we can understand that some may prefer the jitter plots as a matter of personal preference. Therefore, we are including them as supplemental figures, including statistical significance tests. In addition, we have cited the new figures in the discussion to highlight important general trends that are relevant to a system-wide understanding.

Reviewer #2 (Comments for the Author):

This is the second review for this article, which is a follow-up to McGonigle et al. 2019 and explores the functional genes and possible metabolisms in the three uppermost layers of the salt crust in the Bonneville Salt Flats as predicted by shotgun metagenomic data, as well as the capacity for heterotrophy or methanogenesis in aerobic or anerobic (w/ or w/out H₂) conditions using incubation with stable isotope-labeled carbon sources.

Most of the comments from both reviewers have been addressed, with additions to the introduction that more comprehensively describe the importance of this study and the research gap it fills, as well as clarifying which layer is which by referring to the salt pan layers by their mineralogical properties rather than by group number.

However, the main comment that both myself and the other reviewer shared is that, while excellent and comprehensive in itself, the discussion of functional genes/MAGs isn't fully linked to the minerology of the site - i.e., the three different layers - from a systems perspective. Only a few sentences have been added to the discussion and conclusion to address this point, and while the additions are definitely good ones and provide some synthesis, I'm not sure it goes far enough. If, most of the time, there aren't any firm conclusions that can be drawn about microbial

metabolic interplay between the layers (e.g. because, despite differences in abundance, most of these genes are spread throughout the layers), that is fine, but these things are not explicitly stated. (none of this would necessarily be a concern if this journal weren't specifically focused on microbes as they relate to systems). The addition of one or two sentences to other sections of the discussion, similar to the addition of the S-cycling section, and perhaps more specific wording at the beginning of the discussion (how does the abundance of key metabolic genes shift?) would help a lot.

Our Response:

We have added a sentence to the first paragraph of the discussion to highlight specific shifts in key metabolic genes and to cite the new jitter plots requested by Reviewer 1. In addition, we have added sentences throughout the discussion to highlight general trends and differences between layers.

Other minor notes

- Is organic matter as well as microbial biomass concentrated in the gypsum layer? (it certainly looks like it from the picture and from some wording in the text but it's never explicitly stated as far as I can tell)

Our Response:

This certainly appears to be the case from visual inspection, but no measurements of organic content or cellular density were performed during this study.

- Might be interesting to add a sentence speculating on the possibility of how a similar microbial community might operate on a salt flat on ancient Mars, in anerobic conditions

Our Response:

We have added a sentence addressing this point to the conclusion.

- Noticed typos/misspellings line 53, 60.

Our Response:

The typos have been corrected.

October 24, 2022

Dr. William J. Brazelton
University of Utah
257 South 1400 East, Rm. 201
Department of Biology
Salt Lake City, UT 84112-0840

Re: mSystems00846-22R1 (Metabolic Potential of Microbial Communities in the Hypersaline Sediments of the Bonneville Salt Flats)

Dear Dr. William J. Brazelton:

Your manuscript has been accepted, and I am forwarding it to the ASM Journals Department for publication. For your reference, ASM Journals' address is given below. Before it can be scheduled for publication, your manuscript will be checked by the mSystems production staff to make sure that all elements meet the technical requirements for publication. They will contact you if anything needs to be revised before copyediting and production can begin. Otherwise, you will be notified when your proofs are ready to be viewed.

Publication Fees:

If you would like to submit a potential Featured Image, please email a file and a short legend to mSystems@asmusa.org. Please note that we can only consider images that (i) the authors created or own and (ii) have not been previously published. By submitting, you agree that the image can be used under the same terms as the published article. File requirements: square dimensions (4" x 4"), 300 dpi resolution, RGB colorspace, TIF file format.

We recognize that the video files can become quite large, and so to avoid quality loss ASM suggests sending the video file via <https://www.wetransfer.com/>. When you have a final version of the video and the still ready to share, please send it to mSystems staff at mSystems@asmusa.org.

Sincerely,

Sean Gibbons
Editor, mSystems

Journals Department
Supplemental Data Set S2: Accept
Supplemental Figure S2: Accept
Supplemental Figure S1: Accept
Data Set S1: Accept
Figure S3: Accept
Figure S4: Accept